# Embedding Space Selection for Detecting Memorization and Fingerprinting in Generative Models

## Abstract

In the rapidly evolving landscape of artificial intelligence, generative models such as Generative Adversarial Networks (GANs) and Diffusion Models have become cornerstone technologies, driving innovation in diverse fields from art creation to healthcare. Despite their potential, these models face the significant challenge of data memorization, which poses risks to privacy and the integrity of generated content. Among various metrics of memorization detection, our study delves into the memorization scores calculated from encoder layer embeddings, which involves measuring distances between samples in the embedding spaces. Particularly, we find that the memorization scores calculated from layer embeddings of Vision Transformers (ViTs) show an notable trend - the latter (deeper) the layer, the less the memorization measured. It has been found that the memorization scores from the early layers' embeddings are more sensitive to low-level memorization (e.g. colors and simple patterns for an image), while those from the latter layers are more sensitive to high-level memorization (e.g. semantic meaning of an image). We also observe that, for a specific model architecture, its degree of memorization on different levels of information is unique. It can be viewed as an inherent property of the architecture. Building upon this insight, we introduce a unique fingerprinting methodology. This method capitalizes on the unique distributions of the memorization score across different layers of ViTs, providing a novel approach to identifying models involved in generating deepfakes and malicious content. Our approach demonstrates a marked 30% enhancement in identification accuracy over existing baseline methods, offering a more effective tool for combating digital misinformation.

## 1 Introduction

In the rapidly evolving field of artificial intelligence, generative models such as Generative Adversarial Networks (GANs) (Goodfellow et al., 2014) and Diffusion Models (Ho et al., 2020) have emerged as pivotal tools, driving innovation across a myriad of domains including art (Ramesh et al., 2021), media synthesis (Dhariwal & Nichol, 2021), healthcare (Kazerouni et al., 2023), and autonomous systems (Liu et al., 2024). These models possess the remarkable capability to generate new, synthetic data instances that are often indistinguishable from real data, thereby unlocking new frontiers in data augmentation (Trabucco et al., 2023), realistic content creation (Rombach et al., 2022), etc. However, alongside their remarkable capabilities, these models harbor the intrinsic challenge of data memorization.

Data memorization occurs when models inadvertently replicate exact or near-exact pieces of their training data (Somepalli et al., 2023). It presents significant concerns relating to privacy breaches, the integrity of generated content, and the perpetuation of biases ingrained in the training datasets. The nuanced issue of memorization within generative models necessitates robust mechanisms for detecting and quantifying memorization. This mechanism can not only aid in understanding how models learn and replicate data patterns (van den Burg & Williams, 2021) but also in implementing safeguards against misuse, such as unauthorized replication of sensitive information (Wang et al., 2023). Currently, the practice of measuring memorization primarily involves analyzing the embeddings generated from various layers of neural networks, such as Convolutional Neural Networks (CNNs) (Heusel et al., 2018) and Transformers (Pizzi et al., 2022). However, the decision regarding which layer's embeddings should be employed for memorization measurement

has not been formally standardized or thoroughly explored. Common practice often gravitates towards utilizing the embeddings from the penultimate layer of these networks. The lack of justification to the layer choice underscores a pressing gap in the existing methodologies, highlighting the need for a systematic exploration to identify the most effective layers for capturing memorization phenomena accurately. This paper aims to bridge this gap by conducting a comprehensive investigation into the efficacy of memorization measurement using embeddings from different layers of CNNs and transformers. We hope to establish informed guidelines for embedding layer choice, thereby enhancing the ability to correctly quantify the extent of data memorization using the existing methods of measurement.

In this research we study $C_T$-score, an embedding-based data-copying metric proposed in Meehan et al. (2020), to measure model memorization with ViT layer embeddings. Beyond merely selecting the best layers for memorization detection, we introduce and validate a novel approach for model fingerprinting that leverages the unique distribution of the $C_T$-scores layer-wise. The idea comes from the experiment observation, that each model shows a distinctly unique trend in $C_T$-scores calculated by embeddings from different layers. This approach is grounded in the understanding that a model's memorization characteristics are uniquely shaped by its architecture, the data it is trained on, and the optimization techniques used during training. It demonstrates superior performance relative to existing baseline methods in the task of model identification by generated images. This again highlights the vast unexplored possibility of layer-wise study of memorization scores.

## 2    Related Works

Research in data memorization has shown that deep neural networks can memorize random labels due to having significantly more learnable parameters than training examples, indicating a potential to prioritize memorization over generalization (Novak et al., 2018). This observation is complemented by findings from Zhang et al. (2017), which demonstrated that standard regularization strategies are often inadequate in preventing such memorization. The tendency of generative models to memorize training data poses threats to generation quality and integrity and justifies further investigations on both measuring and regularizing their memorizations.

In the realm of generative models, particular attention has been directed towards Generative Adversarial Networks (GANs) and Diffusion Models. Bai et al. (2022) explored methods to reduce memorization in GANs without compromising output quality. Somepalli et al. (2023) investigated strategies to mitigate data copying effects in Diffusion Models. While the techniques for reducing memorizations have been developed, assessment techniques for generative models have also evolved, incorporating metrics like the popular Inception Score (IS), Frechét Inception Distance (FID) (Heusel et al., 2018), Precision and Recall test (Sajjadi et al., 2018), and Self Supervised Copy Detection (SSCD) (Pizzi et al., 2022). Despite their vast diversity, almost all assessment techniques rely on encoded embeddings of generated images, which gives rise to the problem of encoder selection and encoder layer selection. Conventionally, ViT and CNN are used as encoders, and the embeddings are taken from their penultimate layers. While a lack of justification for the embedding layer choice exists, the layer information of ViT and CNN has been studied. Ghiasi et al. (2022) showed that both models behave similarly in the way features progress from abstract patterns in early layers to concrete objects in late layers.

Model fingerprinting aims to create a unique identifier for a machine learning model by analyzing its performance, structure, and behavior to distinguish it from other models. Studies involving model fingerprinting to trace the origins of digital artifacts have grown (Song et al., 2024; Yu et al., 2022). Such efforts are crucial for understanding the source of generative outputs and are instrumental in the battle against the misuse of AI technologies, such as the production of deepfakes. This paper will contribute to fingerprinting generative models by proposing and evaluating a novel method based on memorization detection.

## 3    Observation

Our research uses $C_T$-score as the metric of model memorization. To calculate $C_T$-scores, generated samples are projected into an embedding space, which is often obtained from neural network encoders. Both CNN-

based and transformer-based encoders are used in our initial experiment. The variance of layer-wise $C_T$-score trends resulting from the selection of encoder types is noted. More importantly, an interesting trend is consistently observed when the encoder is transformer-based.

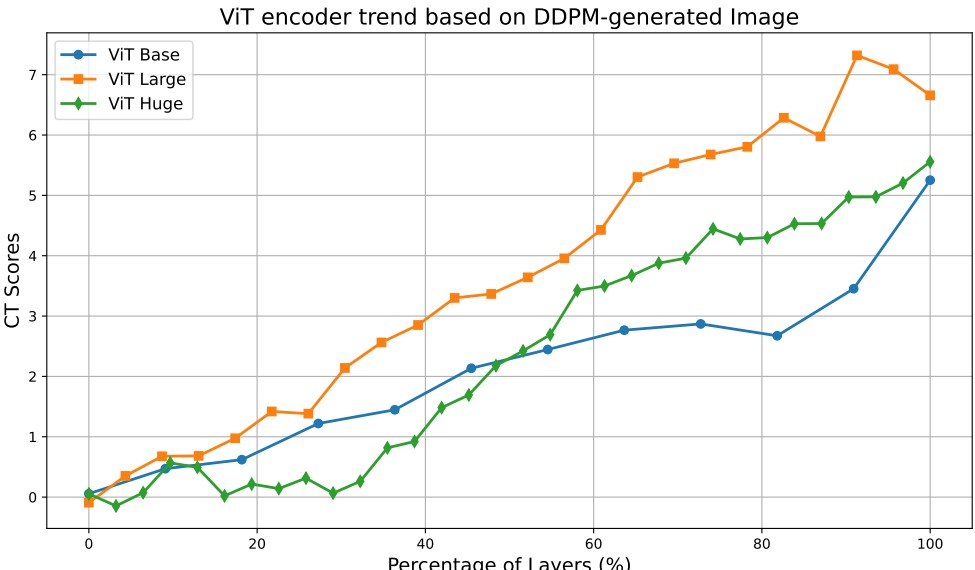

Figure 1: **ViT encoder trend based on DDPM-generated Image**. We use 500 randomly sampled DDPM-generated (Ho et al., 2020) CIFAR-10 Images to compute the $C_T$ scores with three different Vision Transformers, namely "vit-base-patch16", "vit-large-patch16", and "vit-huge-patch14". We observe a consistently increasing trend.

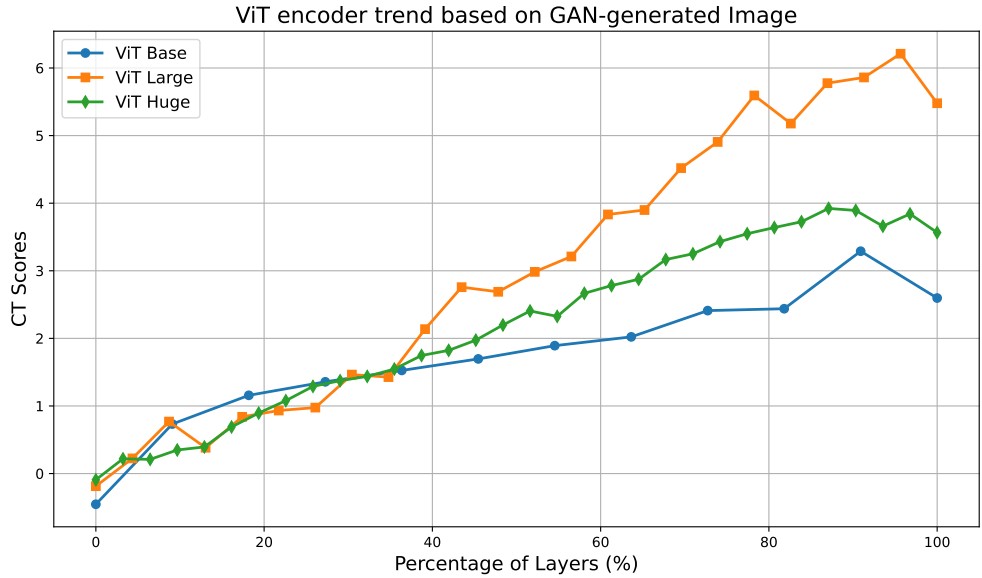

Figure 2: **ViT encoder trend based on GAN-generated Image**. We use 500 random sampled BigGAN-deep (Brock et al., 2019) generated CIFAR-10 Images to compute the $C_T$ scores with three different Vision Transformers, namely "vit-base-patch16", "vit-large-patch16", and "vit-huge-patch14."

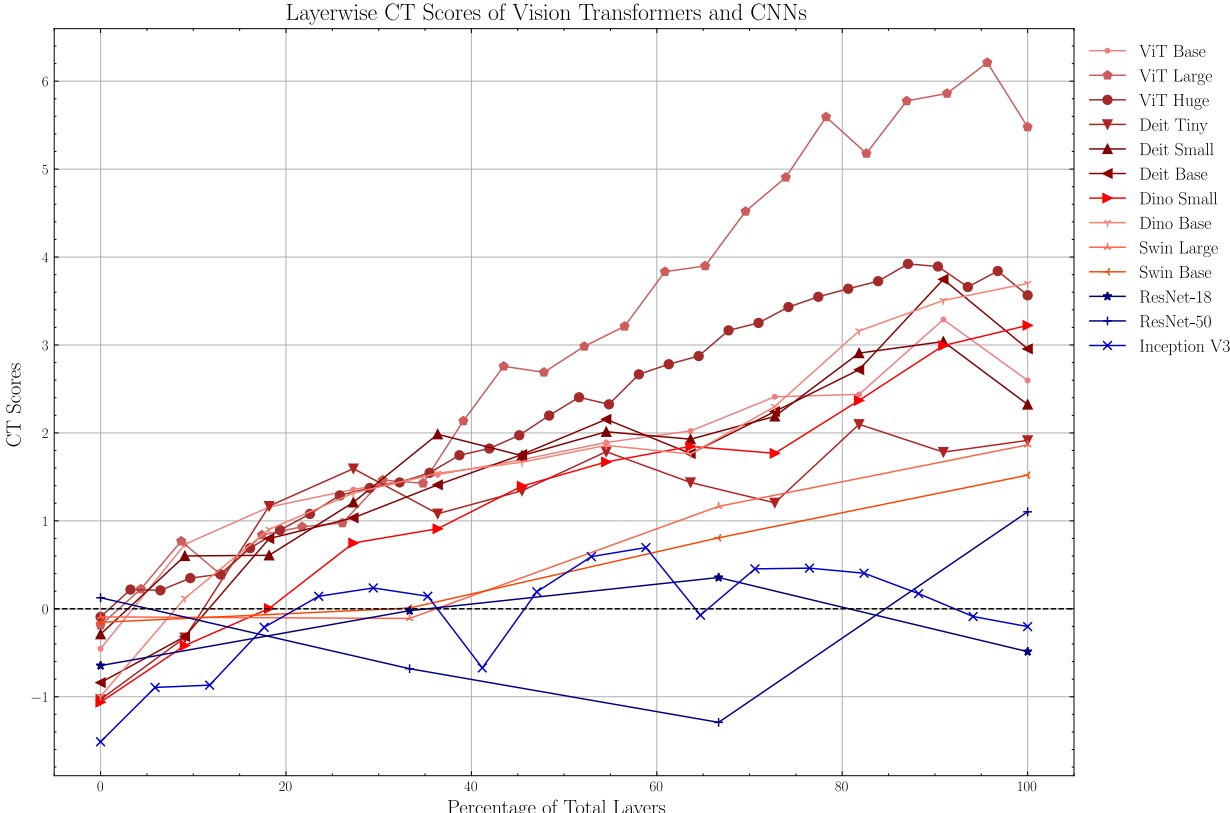

Figure 3: $C_T$ **Score Comparison of ViT based Encoders and CNN Encoders**. This figure illustrates the trends in encoder performance for Vision Transformer, Dino ViT (Caron et al., 2021), DeiT (Touvron et al., 2021), and Swin Transformers (Liu et al., 2021), compared with ResNet (He et al., 2015) and InceptionV3 (Heusel et al., 2018) encoders. Each graph is generated based on 500 GAN generated images, under similar settings to those in Figure 2.

### 3.1 What is $C_T$ Score

The definition of $C_T$ score depends on the concept of data-copying. Meehan et al. (2020) defines that a generative model is data-copying the training set $T$ when in some regions, the model's output distribution $Q$ is systematically closer to $T$ than the true underlying distribution $P$, which $Q$ aims to learn. $C_T$-score is designed as a metric to assess the degree of data-copying by sampling and examining the generated data in relation to the training dataset ($T$) and test datasets (which is seen as samples from $P$). A higher $C_T$ score indicates a lower degree of data-copying and shows the model $Q$'s differentiation from $T$; whereas a lower score indicates a higher degree of data-copying and $Q$'s lack of originality.

Initially, embeddings for the training, test, and generated datasets are computed using pre-trained models, providing high-dimensional representations that capture the essence of the data's features. The embeddings are clustered using the KMeans algorithm, dividing the data embeddings into distinct cells that form a partition $\Pi$ of the instance space $\mathcal{X}$. This step is crucial for assessing data-copying behavior not only globally on the entire $\mathcal{X}$ but also locally within each cell.

Within each cell $\pi$, samples are drawn from the test and generated data embeddings. A distance function $d : \mathcal{X} \to \mathbb{R}$ is needed here to measure how close a sample is to the training set. For all the experiments in this paper, the nearest neighbor distance $d(x) = \min_{t \in T} ||x-t||^2$ is adopted. The samples $P_n$ (drawn from the test dataset) and $Q_m$ (drawn from the generated dataset) are passed into function $d$ and result in $L_\pi(Q_m)$ and $L_\pi(P_n)$, which are considered as samples from the distance distribution. Notice that the distance function

$d$ serves as indicators of similarity, with shorter distances suggesting higher levels of similarity and potential copying. Next, a $Z_U$ score is determined using the Mann-Whitney U test by comparing $L_\pi(Q_m)$ and $L_\pi(P_n)$ and quantifying their statistical difference. A higher $Z_u$ score indicates a greater disparity, implying less similarity and, consequently, less copying. Finally, the $C_T$ score is calculated as a weighted average of the $Z_U$ scores across all cells. The weighting is based on the proportion of the samples in each cell, and cells with an insufficient number of samples are excluded based on a predetermined threshold, ensuring the score only reflects cells with a meaningful representation of generated data. In short, the $C_t$ score can be formulated as

$$C_T(P_n, Q_m) = \frac{\sum_{\pi \in \Pi_\tau} P_n(\pi) Z_U(L_\pi(P_n), L_\pi(Q_m); T)}{\sum_{\pi \in \Pi_\tau} P_n(\pi)}$$

where $P_n(\pi)$ denotes the proportion and $\Pi_\tau$ denotes the set of cells with the number of samples above the threshold $\tau$.

## 3.2 Notable trend for ViT models

The primary objective of our paper is not to improve or justify the CT-score as a method for detecting memorization but rather to explore its behavior across different encoder layers. Using the $C_T$-score formula provided, we conduct initial experiments to calculate layer-wise $C_T$-scores for encoders based on both transformer and CNN architectures. Our findings reveal a unique trend observed in transformer encoders. $C_T$-scores are consistently increasing for latter layers, indicating lower memorization. This phenomenon is especially pronounced in the ViT architecture (as shown in Figure 1, 2, 3 with different ViT implementations). It appears to be a characteristic feature of transformer-based encoders, suggesting a distinct pattern of information memorization associated with the layer hierarchy. *(As a side note, the drop in CT-score in the final layer can be attributed to the alignment of embeddings with classification labels, which compresses the hidden space and reduces the score.)*

In contrast, CNN-based encoders demonstrate a markedly different behavior. Our observations in Figure 3 (details of the encoder pre-training are provided in Appendix A) indicate that CNNs exhibit a relatively consistent or flat $C_T$-score trend across their layers. This uniformity in $C_T$-score distribution suggests that CNN encoders process and represent data in a more homogeneous manner, distinguishing them as stable representers of information when compared to their transformer counterparts. This stability suggests that, in the context of CNN encoders, the choice of layer for embedding extraction might not significantly impact memorization detection, echoing practices seen in other methodologies like Fréchet Inception Distance (FID) (Heusel et al., 2018) where models like InceptionV3 (Szegedy et al., 2015) are preferred for image embeddings extraction due to their consistency.

However, the intriguing results from the ViT models offer a contrasting perspective. Our observation of increasing $C_T$-scores across the layers of transformer encoders prompts a compelling hypothesis that different layers of the ViT model specialize in distinct levels of feature learning. In the context of images, the early layers of ViT concentrate on encoding low-level features, such as simple patterns, colors, and textures. In contrast, the latter layers focus on encoding high-level, more abstract features, such as object categories and complex scene relationships. This hypothesis necessitates a layer-specific analysis on using ViT encoders for embedding extractions, which is key to understanding how the choice of layers influences the memorization detection task.

## 3.3 Low level and high level memorization

We hypothesize that generative models exhibit distinct patterns of memorization at both high and low levels of information, which are crucial for their functioning and output generation. At the low level, these models fall into memorizing basic patterns, features, and textures. This form of memorization represents a limited understanding to elemental aspects such as edges, colors, and simple shapes. On the other hand, high-level memorization involves the integration and synthesis of these basic elements into more complex and abstract concepts. This high-level memorization is not merely a sum of low-level features but represents a more sophisticated semantic understanding and recombination.

In light of the intriguing trend of increasing $C_T$-scores observed in Vision Transformer (ViT) models, our study sought to establish a connection between this trend and the concepts of high-level and low-level memorization. We hypothesize that different layers within the ViT model capture distinct types of memorization representation. Specifically, the embeddings from the initial (front/early) layers would primarily capture low-level memorization, focusing on fundamental features and patterns. The embeddings from the deeper (latter) layers would lean towards high-level memorization, characterized by the integration and synthesis of these basic elements into more complex and abstract representations.

## 4 Experiment

We design an experiment to measure the $C_T$-score trends using a Vision Transformer (ViT) encoder. Five datasets are created from the original CIFAR-10 using various augmentation techniques, which specifically perturb or preserve high-level or low-level information in the images. We expect to see a perturbation reflected in the early or latter parts of the $C_T$-Layer curve when the augmentation targets the low-level or high-level information, respectively. We use a BigGAN-deep model as the baseline model and the $C_T$-Layer curve calculated from its generation as the baseline curve to be compared against.

### 4.1 Dataset preparation

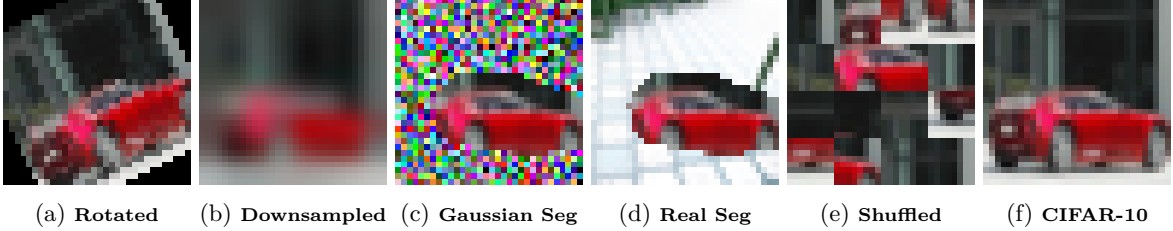

(a) **Rotated**  (b) **Downsampled** (c) **Gaussian Seg**  (d) **Real Seg**  (e) **Shuffled**  (f) **CIFAR-10**

Figure 4: Samples for curated datasets

We curate five distinct datasets (examples: Figure 4), which are augmented versions of CIFAR-10 to delve into the intricacies of low-level and high-level data processing mechanisms. Each modified version of the CIFAR-10 dataset (Figs. 4a-4e) is treated as an output from "pathological generative models," memorizing either low-level or high-level information disproportionately. Our objective is to ascertain whether modifications to either low-level or high-level features precipitate discernible alterations in the $C_T$-Layer curve, indicating the effect of such modifications. The deliberate modification of the CIFAR-10 dataset results in a high average dissimilarity between the augmented and original datasets. Consequently, the $C_T$-scores predominantly indicate underfitting (high positive values). Nevertheless, the relative trend observed in the $C_T$-Layer curve remains insightful as we are only interested in seeing shifts from the baseline consistently-increasing trend.

For the analysis of low-level features, we construct two datasets (Figs.4a-4b). We aim for these two datasets to imitate the generation of models that primarily alter low-level details while preserving high-level semantics. The first dataset (4a) is generated through the application of rotation-based modifications to CIFAR-10 images, incorporating a spectrum of angular orientations. The second dataset (4b) is constructed from the process of downsampling the original images followed by upsampling them back to their original dimensions.

Although the first two datasets intend to only perturb the low-level semantics, the processes of rotation and down-upsampling still corrupt the objects in the images to a degree, affecting the high-level information. To address this issue, the subsequent two datasets are crafted to imitate the generation of models that extract and preserve the training image's high-level semantics. In our exploration of such data processing techniques, we employ a fine-tuned ResNet-18 model (details of the fine-tuning process are provided in Appendix B) to conduct image segmentation on images from the CIFAR-10 dataset. This segmentation isolates the foreground elements of the images, which are then seamlessly integrated onto alternative backgrounds. Specifically, the modification entails replacing the original image backgrounds with Gaussian noise 4c and real-world landscapes 4d, sourced from the BG-20k (Li et al., 2021) dataset. This strategic modification is

designed to ensure that the fundamental semantics of the image—primarily its main object—remain largely unaffected.

Following the four datasets that preserve high-level information while altering low-level details, we construct an additional dataset (4e) that largely preserves the low-level details while significantly alters the high-level information. This dataset is generated by fragmenting each original CIFAR-10 image into 16 equal-sized square patches and shuffling them. Each image in this dataset has the exact collection of pixels, though arranged differently, as its original counterpart. This can be viewed as an intentional preservation of the low-level image features since color and pattern details within each piece are exactly the same as in the original.

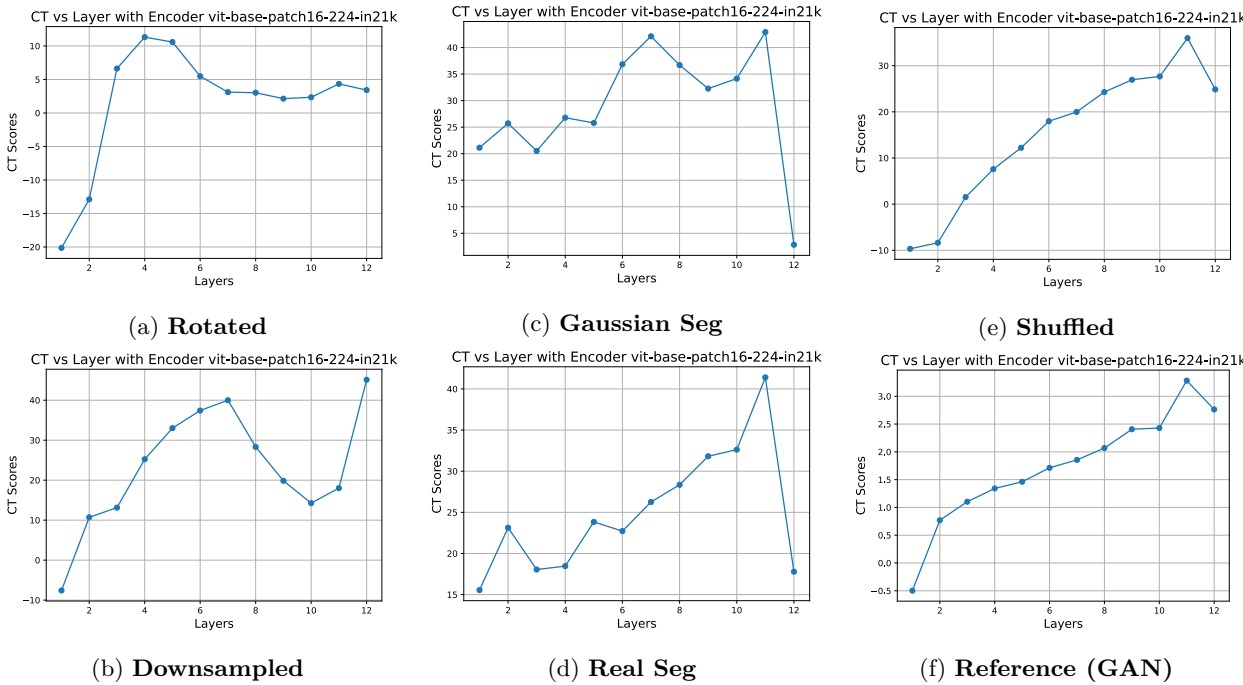

Figure 5: $C_T$ vs Layers Curve for Six different dataset

## 4.2 $C_T$-**Scores Analysis**

We employ a non-pathological generative model (specifically, BigGAN-deep model in Fig.2 ) as a baseline model, and use a ViT-base model as the encoder to derive the $C_T$-layer curves for all six models (Fig.5). The observed shifts of the augmented dataset $C_T$-layer trends from the baseline curve closely align with our hypothesis in 3.3 that embeddings from initial layers capture low-level memorization, while those from deeper layers exhibit high-level memorization. The shifts can be reflected by deviations from the baseline, roughly monotonically increasing curve in the form of peaks and valleys at different locations.

For datasets augmented with low-level detail modifications – rotation and resolution changes (5a, 5b) – we observe a significant shift in their $C_T$-layer trends at the initial layers, compared with the baseline increasing trend (5f). The $C_T$-layer curves for both rotation and down-upsampling display an early peak at the initial layers. Since these two datasets are more dissimilar to the original CIFAR-10 in terms of low-level details, this observation underscores the sensitivity of the network's earlier layers to alterations in low-level image details.

On the other hand, the analysis of high-level detail preservation, achieved through the alteration of image backgrounds while preserving the foreground elements, reveals the latter network layers' focus on high-level semantic information. There are discernible drops of the relative $C_T$-scores on the last layer of the network for the two segmentation methods (5c, 5d), which both intend to cause overfitting on high-level information by

preserving the semantics of original CIFAR-10. This drop suggests that the final layers prioritize classification decisions based on high-level semantics, as expected, where the preservation of high-level features leads to lower memorization tendencies.

In the analysis of the $C_T$-scores curve for the shuffled dataset (5e), the observed deviation is relatively subtle, attributable to the inherently increasing trend of the baseline reference. Consequently, even an enhancement in the scores of the deeper layers aligns with this prevailing trend. However, this subtle increase still indicates a diminished capacity to differentiate between meaningful semantic information due to the shuffling process. Notably, the stability in the score of the front layers, despite significant alterations to high-level details, substantiates the hypothesis that the deeper layers are chiefly tasked with the extraction of high-level semantic information from the input images.

### 4.3 Further Experiment

Our first experiment analyzes layer-wise $C_T$-scores trends on reflecting the memorization of low-level and high-level information. It supports our hypothesis in 3.3 on the difference in memorization detection between different layers of encoder embeddings, although heavily relying on the assumption that all the specially augmented images are samples from some imaginary "pathological generative models". Beyond that, it builds upon the original hypothesis, proposing that the $C_T$-scores patterns can be indicative of the generative models' memorization natures. Our further experiment aims to verify this point by training and sampling from real generative models of two specific architectures - denoising diffusion probabilistic models (DDPM) (Ho et al., 2020) and denoising diffusion implicit models (DDIM) (Song et al., 2022). DDPM and and DDIM are both Diffusion Models that generate data distributions from noises by adding noise to data over time steps and then learning to reverse the process. DDPM follows strict Markovian assumption at inference time, while DDIM follows a deterministic non-Markovian formulation. Unlike our previous experiment where the memorization nature (either low or high) of the augmented images is artificially crafted and known, it's impossible to verify if the $C_T$-score trends reflect the memorization nature of the diffusion models and thus are indicative patterns since the memorization natures of DDPM and DDIM are unknown to us. Hence, we adopt a similarity comparison method by training multiple checkpoints of each of the two architectures, calculating a $C_T$-Layer curve per checkpoint, and comparing the cosine similarities between $C_T$-Layer curves of each pair of checkpoints, from the same (intra-) or different (inter-) architectures.

Multiple checkpoints are taken at several epochs for both DDPM and DDIM (details of the training are provided in Appendix C). To ensure the checkpoints can yield meaningful generations, only checkpoints at epochs after the loss function largely converges are recorded to ensure the image generation is meaningful. In this case, epoch number 120, 150, 180, 210, 240, and 270 are used. The layer-wise $C_T$-Layer scores curves 6a show a similar increasing trends for both architectures on all checkpoints. This meets our expectation because it's unlikely that the added number of training epochs will change the model memorization natures drastically. The distinction between DDPM and DDIM architectures might not be prominent due to the fact that they adopt very similar diffusion methods.

However, the subtle divergence between the overall DDPM trend (orange curves) and DDIM trend (blue curves) is still noticeable. This makes it intriguing to compare the intra-architecture (e.g. DDPM180 and DDPM210) and inter-architecture (e.g. DDPM180 and DDIM180) $C_T$-Layer curve similarities. By treating each $C_T$-Layer curve as a vector in $\mathbb{R}^{11}$, where 11 is the number of encoder layers, the cosine similarity between each pair of $C_T$-Layer curves is computed and recorded in a heat map 6b. The overall intra-architecture similarity is higher than the inter-architecture similarity, even if the raw curves in 6a look fairly similar. This experiment result proves that $C_T$-Layer patterns are indicative of the architectures of generative models by displaying marked differences between measurements from different architectures and are sufficiently robust to minor differences between checkpoints of the same architecture.

## 5 Fingerprinting

Our findings indicate that the $C_T$-Layer curve serves as a meaningful indicator of the model's handling of different levels of details, suggesting that this curve could act as a distinctive feature reflective of the model.

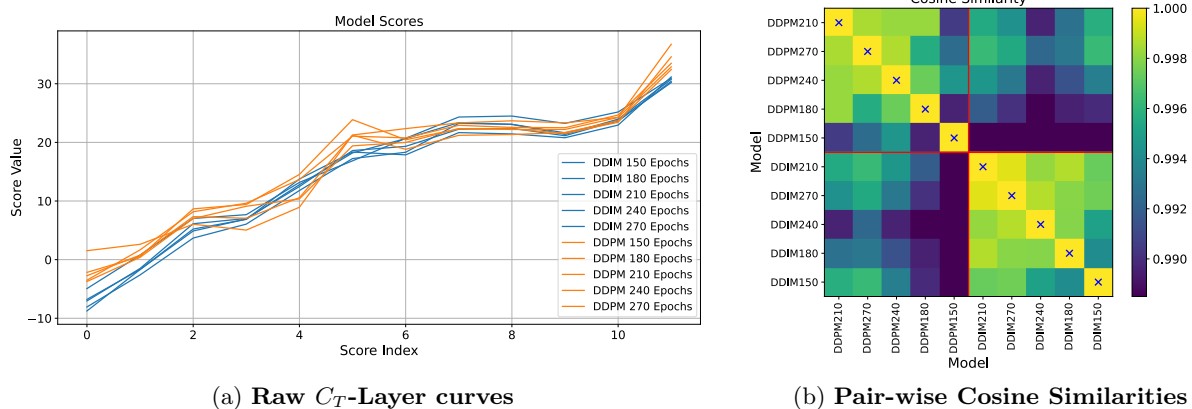

(a) **Raw $C_T$-Layer curves**    (b) **Pair-wise Cosine Similarities**

Figure 6: $C_T$**-scores of Multiple Checkpoints of DDPM and DDIM** (a) The raw $C_T$-Layer curves of checkpoints when training epochs = 150, 180, 210, 240, 270 for both DDPM and DDIM (b) The heat map records all pair-wise cosine similarities between checkpoints. The heat map is symmetric, with the 1st and 3rd quadrants displaying inter-architecture similarities, and the 2nd and 4th quadrants displaying intra-architecture similarities. The diagonal is crossed out since diagonal blocks correspond to all the checkpoints' similarities with themselves (always 1).

This insight paves the way for utilizing the $C_T$-Layer curve as a technique for fingerprinting generative models, an application growing in importance within the field as people are more and more worried about deepfake and malicious use of generated media.

## 5.1 Introduction to Model Fingerprinting Using $C_T$-Layer Curves

Traditional methods of fingerprinting generated images often rely on access to the processed training images (Yu et al., 2022) utilized by the model during its training phase, a requirement that is not always practical or feasible. In contrast, our proposed methodology offers a novel approach to extract characteristic features of a given model architecture by analyzing its corresponding $C_T$-Layer scores 7. For the purpose of this study, we employ the ViT-base model as our encoder; However, the methodology is adaptable and could potentially be extended to other variations of ViT encoders.

In more detail, our method involves calculating the $C_T$ score for each layer of the target image set. Subsequently, we employ these scores in conjunction with L2-norm and cosine-similarity nearest neighbor analysis to identify the closest match within our dataset. This dataset comprises $C_T$-Layer scores for a diverse array of known models, enabling the identification and matching process. Through this approach, we introduce a robust mechanism for model fingerprinting that bypasses the need for direct access to a model's training images, thereby offering a more versatile and accessible method for characterizing and identifying generative models.

## 5.2 Experimental Setup for Evaluating Fingerprinting Accuracy

To empirically validate our proposed fingerprinting methodology, we conduct a preliminary experiment encompassing a diverse set of 18 model samples derived from various generative model architectures. The architectures are ContraGAN, BigGAN, and SNGAN for GANs and DDPM, DDIM, and PNDM for Diffusion Models. For each architecture, we have three model samples, each generating 500 images, resulting in a total of 1,500 generated images per architecture.

To establish a comparative framework, we implement a baseline method that utilizes the mean, standard deviation (std), and Fréchet Inception Distance (FID) score to extract features from the generated images and perform nearest-neighbor matching. Additionally, we compare our approach with several neural network classification methods, including a vanilla CNN model trained from scratch, a fine-tuned ViT-base model,

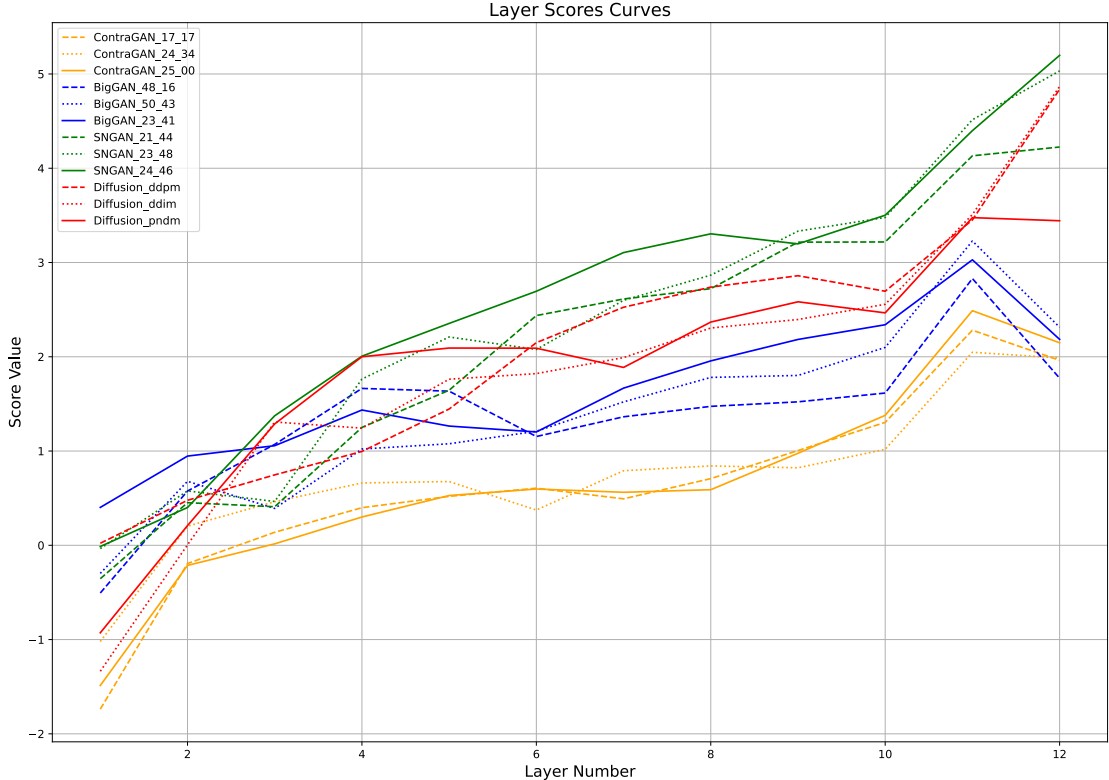

Figure 7: **Differentiation of Model Architectures Utilizing $C_T$-Layer Scores Curves.** The displayed graph highlights distinct trajectories for each model, revealing the utility of $C_T$ Layers curves in differentiating between various model architectures. The convergence of models with the same architecture into identifiable clusters further reinforces the curves' efficacy as a discriminative tool. This demonstrates the curves' potential as a reliable method for architectural distinction and analysis in model evaluation.

and a fine-tuned ResNet50 model (details of the training and fine-tuning process are provided in Appendix D).

Our experimental setup is meticulously crafted to evaluate the accuracy of our method. From the 1,500 generated images for each model architecture, we first separate 500 images as the testing set. The remaining 1,000 images are then used to form two training datasets (Datasets_small with 500 images and Datasets_full with the entire 1,000 images) to create ablation studies. For our $C_T$ method, we formulate the training-set $C_T$-scores for each model using its corresponding training images. At testing time, for each target, we calculate its $C_T$-score using the testing set and identify the best match in the training-set $C_T$-scores with different matching methods. The overall accuracy is determined by the proportion of correctly matched target models. For the baseline method, the overall process mirrors that of the $C_T$ method, but we use baseline statistical scores instead.

We train different neural networks on training sets and evaluate their accuracy on the testing set. We use majority votes for final classification since each model takes a single image as the input, while our $C_T$ and baseline methods are applied to the entire testing dataset. By employing this systematic approach, we aim to provide a thorough and robust assessment of our proposed method's performance relative to both statistics-based and neural network-based techniques.

| Method | Dataset (small) | | Dataset (full) | |
|---|---|---|---|---|
| | **Accuracy** | **Time(s)** | **Accuracy** | **Time(s)** |
| Baseline Method (L2 Norm) | 0.666 | **22.46** | 0.833 | **44.73** |
| Baseline Method (Cosine Similarity) | 0.333 | **22.46** | 0.333 | **44.73** |
| Baseline Method (L2 Norm + Cosine Similarity) | 0.666 | **22.46** | 0.833 | **44.73** |
| Vanilla CNN Model | 0.500 | 200 | 0.833 | 400 |
| ViT-base Model | 0.167 | 1830 | 0.333 | 3642 |
| ResNet50 Model | **0.833** | 1130 | **1.000** | 2278 |
| **Our:** $C_T$-Layer Method (L2 Norm) | 0.666 | 1417 | **1.000** | 2835 |
| **Our:** $C_T$-Layer Method (Cosine Similarity) | 0.666 | 1417 | 0.833 | 2835 |
| **Our:** $C_T$-Layer Method (L2 Norm + Cosine Similarity) | **0.833** | 1417 | **1.000** | 2835 |

Table 1: Comparison of Baseline Model and $C_T$-Layer Method with Time Metrics

### 5.3 Results and Performance Analysis

The results outlined in Table 1 underscore the effectiveness of our proposed $C_T$-layer methodology. Compared to the baseline methods, our approach offers a significant improvement in performance, demonstrating its capability to capture and leverage architectural nuances for model fingerprinting. The $C_T$-Layer Method records a remarkable increase in accuracy for both the full and small sets, with an outstanding 100% accuracy in the full set and an 83.3% accuracy in the small set using the combination of L2 Norm and Cosine Similarity. This performance significantly outstrips the baseline methods, which peak at 66.6% and 83.3% accuracy for similar conditions.

Additionally, our methodology outperforms conventional models such as Vanilla CNN and ViT-base, and it achieves comparable results to ResNet50, while without the need for training. ResNet50 is pre-trained on ImageNet and performs well on datasets similar to ImageNet, such as CIFAR-10, its performance may degrade on datasets that deviate significantly from ImageNet in terms of distribution and content. In contrast, our method is unaffected by changes in dataset characteristics and can be generalized better in such cases. Although our method employs a ViT-base model as the encoder, direct finetuning a ViT-base model to learn fingerprinting classification yields suboptimal results due to limited data availability — a common challenge in fingerprinting scenarios.

Moreover, we evaluate the training duration of our method against other models. The baseline method is exceptionally quick due to its simplistic statistical nature. However, our approach demonstrates comparable processing times to ResNet50 and is faster than the ViT-base model, underscoring its efficiency.

These results suggest that the $C_T$-layer scores, by leveraging layer-specific responses to various image details, provide an effective method for model identification and characterization. This improvement is critical in scenarios where precision and reliability in model fingerprinting are paramount. That the $C_T$-Layer Method outperforms the established models confirms its efficacy and potential for generative model fingerprinting.

## 6 Conclusion

This study pioneers an approach to understanding data memorization in generative models, providing a detailed analysis of the impact of layer selection in Convolutional Neural Networks (CNNs) and Transformers on memorization detection. It reveals distinct memorization patterns between CNNs and Vision Transformers (ViTs), underscoring the importance of layer-specific analysis for accurate memorization quantification. Furthermore, the introduction of a novel model fingerprinting technique leveraging $C_T$-score distributions marks a significant advancement in identifying generative models, offering potential tools for the ethical use of generative models, and addressing critical concerns around privacy, content integrity, and the proliferation of deepfakes.

**Limitations and future directions** While our proposed technique offers significant advancements, several practical limitations and challenges need to be considered for real-world applications.

Due to computational constraints, we focused on CIFAR-10, but we believe the trends observed in model architectures, particularly CT-scores across layers, remain valid for larger datasets as well. The choice of layer for memorization detection varies depending on the type of memorization (low-level vs. high-level), which can be challenging to distinguish without extensive analysis. Future work should develop hybrid detection methods that analyze both low-level and high-level features and automate layer selection algorithms for better accuracy. Although our fingerprinting method does not require direct access to the model's training dataset, it still necessitates a general, compact, and sufficiently generic dataset to serve as a practical baseline. Future directions should include curating such datasets and implementing dynamic updates to include new and emerging generative model characteristics.

Our method requires pre-computed $C_T$-Layer score data for various generative model architectures. Future efforts should focus on expanding this database to include new models and establishing collaborations with developers for early access. The computational complexity of computing $C_T$-Layer scores for each layer can be intensive. Future research should aim to optimize computation processes and leverage high-performance computing resources to improve efficiency and scalability.

Implementing our fingerprinting technique requires careful consideration of ethical and privacy implications. Establishing clear guidelines and ethical standards for deployment, ensuring data privacy, and developing protocols for transparency and accountability will be crucial. Collaborations with ethicists, policymakers, and industry stakeholders will help align technological advancements with societal values and norms.

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

## Appendix

### A    Encoder Pretraining Details

In this section, we provide the pretraining details for the encoders used in our experiments. All encoders were pretrained on various versions of the ImageNet dataset, which is widely used in the research community for model pretraining due to its large-scale nature and diverse categories. The specific models and their pretraining sources are detailed below:

- **ViT-base-patch16**: Pre-trained on ImageNet-21k (14 million images, 21,843 classes) at resolution 224x224, and fine-tuned on ImageNet 2012 (1 million images, 1,000 classes) at resolution 224x224. This transformer-based encoder was accessed via HuggingFace. `https://huggingface.co/google/vit-base-patch16-224`

- **ViT-large-patch16**: Similar to the ViT-base model, but with a larger number of parameters, also pretrained on ImageNet-21k (14 million images, 21,843 classes) at resolution 224x224, and fine-tuned on ImageNet 2012 (1 million images, 1,000 classes) at resolution 224x224. `https://huggingface.co/google/vit-large-patch16-224`

- **ViT-huge-patch14**: A larger version of the ViT architecture, also pretrained on ImageNet-21k (14 million images, 21,843 classes) at resolution 224x224, and fine-tuned on ImageNet 2012 (1 million images, 1,000 classes) at resolution 224x224. `https://huggingface.co/google/vit-huge-patch14-224-in21k`

- **Deit-tiny-patch16**: A data-efficient image transformer (DeiT), pretrained and fine-tuned on a large collection of images in a supervised fashion, namely ImageNet-1k, at a resolution of 224x224 pixels. `https://huggingface.co/facebook/deit-tiny-patch16-224`

- **Deit-small-patch16**: Another transformer-based encoder from the DeiT family, pretrained and fine-tuned on a large collection of images in a supervised fashion, namely ImageNet-1k, at a resolution of 224x224 pixels. `https://huggingface.co/facebook/deit-small-patch16-224`

- **Deit-base-patch16**: The larger version of the DeiT model, also pretrained and fine-tuned on a large collection of images in a supervised fashion, namely ImageNet-1k, at a resolution of 224x224 pixels. `https://huggingface.co/facebook/deit-base-patch16-224`

- **ResNet-18**: A smaller convolutional neural network (CNN) architecture trained on imagenet-1k. `https://huggingface.co/microsoft/resnet-18`

- **ResNet-50**: A widely-used CNN architecture pre-trained on ImageNet-1k at resolution 224x224, known for its effectiveness in image classification tasks. `https://huggingface.co/microsoft/resnet-50`

- **Inception-v3**: A CNN architecture designed for efficient computation, pretrained on ImageNet-1k. `https://huggingface.co/timm/inception_v3.tv_in1k`

All the models were pretrained on ImageNet with image resolution sizes that are consistent with the model architectures. The use of ImageNet pretraining is crucial for ensuring strong performance across a wide range of tasks. However, we acknowledge that dependence on ImageNet may have implications for how well the models generalize to datasets with different characteristics. Despite this, our method aims to reduce the reliance on pretraining by leveraging layer-wise representations for fingerprinting tasks, which we believe makes it robust across different domains.

## B  Fine-Tuning the ResNet-18 Model for Image Segmentation

This section provides details on the fine-tuning process of the ResNet-18 model for image segmentation tasks on the CIFAR-10 dataset.

We start with a pre-trained ResNet-18 model from PyTorch Paszke et al. (2019) and modify its architecture to suit the image segmentation task based on Lehman (2019)'s implementation. The first convolutional layer is replaced to adapt to CIFAR-10 image dimensions. The final fully connected layers are removed, and a bilinear upsampling layer is added to increase the resolution of feature maps. Additionally, a convolutional layer for classification with 10 output channels (corresponding to the CIFAR-10 classes) is appended, followed by another bilinear upsampling layer to match the output size to the input size. The CIFAR-10 dataset is used for both training and testing. For training, the images are transformed using random cropping, horizontal flipping, normalization, and conversion to tensors. For testing, the images are normalized and converted to tensors without any augmentation.

The training procedure involves setting up the loss function, optimizer, and learning rate scheduler. The model is trained for 30 epochs with the following details:

- **Loss Function:** Binary Cross-Entropy with Logits Loss (BCEWithLogitsLoss)
- **Optimizer:** Stochastic Gradient Descent (SGD) with a learning rate of 0.05, momentum of 0.9, and weight decay of 1e-4
- **Learning Rate Scheduler:** Cosine Annealing with a minimum learning rate of 0.001
- **Batch Size:** 128 for training, 100 for testing
- **Number of Epochs:** 30
- **Number of Workers:** 16 for data loading

The training and evaluation of the model are conducted on a machine equipped with an NVIDIA GeForce GTX 1080 Ti GPU. This computational environment facilitates the efficient processing of the CIFAR-10 dataset and the fine-tuning of the ResNet-18 model.

During training, both training and validation loss and accuracy are monitored to evaluate the model's performance. These metrics are plotted to visualize the model's learning progress over the epochs. The segmentation results are also visualized by converting the segmented outputs to images, highlighting the model's ability to segment different classes within the CIFAR-10 dataset.

| Method | DDPM | DDIM |
|---|---|---|
| **Noise-predicting Model Architecture** | Unet2D | |
| **Loss Function** | MSE Loss | L1 Loss |
| **Training Time Step** | 1000 | |
| **Inference Time Step** | 1000 | 250 |
| **Training Batch Size** | 16 | |
| **Optimizer** | AdamW (LR: 0.0001) | |
| **Device** | NVIDIA GeForce GTX 1080 Ti GPU | |
| **Training Dataset** | CIFAR-10 (Resolution: 32) | |

Table 2: Summary of DDPM and DDIM Method Configurations and Training Settings

## C  Training DDIM and DDPM Models

This section details training DDIM and DDPM models and collecting their checkpoints at multiple epochs.

The DDPM model implementation follows the original DDPM paper (Ho et al., 2020). We use the official training and inference code posted by the authors on the Hugging Face website. The DDIM implementation

follows a GitHub Repository: Basara (2022). Table 2 summarizes the details of training and inference configurations of the two methods.

## D Baseline Neural Network Models for Fingerprinting Performance Comparison

This section provides details on the baseline neural network models used for performance comparison in the context of fingerprinting tasks. Specifically, we describe the training of a vanilla Convolutional Neural Network (CNN) model, a fine-tuned ResNet-50 model, and a fine-tuned Vision Transformer (ViT) model from PyTorch Paszke et al. (2019).

The table below summarizes the architecture, loss function, optimizer, and other training settings for the three models:

| Model | Vanilla CNN | ResNet-50 | Vision Transformer (ViT) |
|---|---|---|---|
| **Architecture** | 3 Conv layers, 2 FC layers | Pre-trained, modified FC layer | Pre-trained, modified head |
| **Loss Function** | Cross-Entropy Loss | | |
| **Optimizer** | Adam (LR: 0.001) | Adam (LR: 0.001) | Adam (LR: 0.01) |
| **Epochs** | 50 | 50 | 30 |
| **Batch Size** | 32 | 32 | 32 |
| **Device** | NVIDIA GeForce GTX 1080 Ti GPU | | |
| **Dataset** | Fingerprinting dataset with resize to 224x224 (ResNet-50 and ViT) | | |
| **Transforms** | Resize, Normalize | | |

Table 3: Summary of Model Architectures and Training Settings for Fingerprinting

The performance of the models is evaluated using accuracy metrics. During training, the average loss and accuracy per epoch are monitored. After training, a majority vote mechanism is used for evaluation on the test dataset. Predictions for each label are collected, and the most common prediction is considered as the final prediction for that label. The accuracy is calculated based on the correct majority votes, providing insight into the models' classification capabilities for the fingerprinting task.

