# OpenReview forum: "Embedding Space Selection for Detecting Memorization and Fingerprinting in Generative Models"
_TMLR — Rejected by TMLR_

### Review · Reviewer_Sf4b · 2024-08-21

**Summary Of Contributions:**

The paper presents an investigation into the memorization process in generative models, such as Generative Adversarial Networks (GANs) and Diffusion Models (DMs). The authors approach this problem by analyzing the features of pretrained encoders, which serve as a proxy for understanding the memorization behavior of these generative models.

Through their analysis, the authors establish a correlation between the depth of the encoder layers and the degree of memorization exhibited by the model. This finding provides insights into the hierarchical nature of the learned representations within convolutional neural networks (CNNs) and vision transformers (ViTs) used in the generative models.

Building upon these observations, the authors design a fingerprinting mechanism that can be used to identify the generative models.

**Audience:**

Yes

**Broader Impact Concerns:**

The authors already discuss ethical concerns in the Conclusion section.

**Claims And Evidence:**

No

**Requested Changes:**

- Provide a detailed description of the encoder pretraining, including the dataset used and the training procedure, to allow for a better understanding of the potential impact on downstream applications.
- Expand the experimental evaluation to include larger and more diverse datasets, such as Stable Diffusion's training data, to assess the generalizability of the observed memorization behavior. I recognize this point might be difficult and costly to run, other possibilities include smaller, but still larger than CIFAR-10, datasets, such as ImageNet.
- Provide a clear justification for the use of multiple models at different training epochs, and further elaborate on the interpretation of the results shown in Figure 6(b).
- Discuss the potential implications of the perfect accuracy on the fingerprint identification task.

**Strengths And Weaknesses:**

### Strengths

- The paper proposes an interesting analysis of the memorization and fingerprinting of generative models on CIFAR10 across several encoders.

---

### Weaknesses

- The authors do not provide sufficient details regarding the training of the encoders used in their experiments. This lack of information makes it difficult to assess the potential impact of the pretraining dataset on the downstream applications explored in the paper.
- While the authors mention the hierarchical nature of the learned representations in convolutional neural networks (CNNs) and vision transformers (ViTs), they do not adequately position their work with respect to prior research on this topic, such as the work by Ghiazi and Kazemi.
- The related work section of the paper is unsatisfactory, as it reads more like a list of previous approaches rather than a thoughtful discussion of how the current work builds upon or differs from these prior efforts.
- The authors make claims about the models' potential ability to recognize and generate faces or landscapes ("In visual models, this could mean recognizing and generating faces or landscapes" in Section 3.3), but these claims are not supported by the experimental evidence provided in the paper. It would be important to either substantiate these assertions or refrain from making such broad implications.
- Evaluating the models solely on the CIFAR-10 dataset may not be sufficient, as this dataset is relatively small and can be easily overparameterized. The authors should consider expanding their analysis to larger and more diverse datasets, such as Stable Diffusion's training data (LAION), to assess the generalizability of the observed memorization behavior.
- The rationale for using multiple models at different training epochs, as described in Section 4.3, is not clearly explained. It would be helpful if the authors could provide a more detailed justification for this approach and further elaborate on the interpretation of the results shown in Figure 6(b), as it shows a range from 0.99 to 1., and the lower values are obtained with the undertrained models.
- The authors' approach appears to achieve perfect accuracy on the fingerprint identification task, which raises questions about potential data leakage or the inherent simplicity of the task. Additionally, the fact that a ResNet-50 model can achieve the same level of accuracy with a reduced running time raises the question of the practical advantages of the authors' method. A more in-depth discussion of these aspects would strengthen the paper's contribution.

---

Ghiasi, A., Kazemi, H., Borgnia, E., Reich, S., Shu, M., Goldblum, M., Wilson, A.G. and Goldstein, T., 2022. What do vision transformers learn? a visual exploration. arXiv preprint arXiv:2212.06727.

---

> ### Author Response · Authors · 2024-10-03
> **Response to Reviewer Sf4b part 1**
>
> We thank the reviewer for the detailed and insightful feedback. In response to the comments, we have addressed each point below:
>
> ---
>
> ## Weakness 1 / Requested Change 1: Encoder Pretraining Details
>
> **Comment:**
> The reviewer raised concerns regarding the lack of details about the pretraining of encoders and how the pretraining dataset might affect downstream tasks.
>
> **Response:**
> Thank you for pointing this out. We agree that providing more information about the encoder pretraining is essential for better understanding the impact on downstream tasks. All encoders used in our experiments were pretrained on variants of the ImageNet dataset. For instance, ViT models such as `vit-base-patch16`, `vit-large-patch16`, and `vit-huge-patch14` were pretrained using the training procedures provided by HuggingFace. Similarly, ResNet-18, ResNet-50, and Deit models follow ImageNet-based pretraining as well.
>
> We will include a detailed description of the pretraining process in Section 3.2 and the Appendix, where we will outline the exact datasets and training parameters used for the encoders. While ImageNet pretraining has an impact on downstream performance, it is not specific to our work, and the broader conclusions we draw are not limited by this particular pretraining dataset. We will clarify this in the manuscript.
>
> ---
>
> ## Weakness 2 & 3: Prior Work and Related Work Section
>
> **Comment:**
> The reviewer mentioned that the related work section reads like a list of prior approaches and lacks discussion on how the current work builds upon or differs from these efforts.
>
> **Response:**
> We acknowledge this concern and will revise the related work section to provide a more thoughtful discussion on how our work compares to and builds upon prior research, including the work of Ghiazi and Kazemi. This will include more detailed citations and explanations of how our approach differs from and extends these works, particularly in terms of how we address fingerprinting and memorization detection in generative models.
>
> ---
>
> ## Weakness 4: Unsubstantiated Claims in Section 3.3
>
> **Comment:**
> The reviewer pointed out that our claims about models recognizing and generating faces or landscapes (Section 3.3) are not backed by experimental evidence.
>
> **Response:**
> We appreciate this feedback. We agree that these claims need to be either substantiated or reframed. We will clarify this in the manuscript.
>
> ---
>
> ## Weakness 5 / Requested Change 2: CIFAR-10 Dataset Limitation
>
> **Comment:**
> The reviewer raised concerns about using only CIFAR-10 for evaluation, suggesting that the dataset may be too small and easily overparameterized. They recommended expanding the analysis to larger datasets like ImageNet or LAION.
>
> **Response:**
> We acknowledge the concern regarding the use of CIFAR-10, a smaller dataset that might raise questions about the generalizability of our findings. While larger datasets such as ImageNet or LAION would provide more diversity, due to computational constraints, we focused our analysis on CIFAR-10 as it allows us to efficiently evaluate the models within our available resources.
> That being said, we believe the core insights drawn from this study, particularly the behavior of CT-scores across model layers, remain valid and generalizable. The layer-wise representation and memorization trends we observe are based on model architectures (ViTs, CNNs) and their processing of hierarchical representations, which we anticipate would apply to larger datasets as well. We will clarify this generalizability in the paper while acknowledging the computational limitations that guided our dataset choice.

---

> > ### Author Response · Authors · 2024-10-03
> > **Response to Reviewer Sf4b part 2**
> >
> > ## Weakness 6 /  Requested Change 3: Use of Multiple Models at Different Epochs
> >
> > **Comment:**
> > The reviewer requested further justification for using models at different training epochs and clarification on the interpretation of the results in Figure 6(b).
> >
> > **Response:**
> > We acknowledge the insufficient explanation provided in the first draft. We revised Section 4.3 to emphasize the further experiment’s goal of proving CT-layer patterns can be indicative of the generative models' memorization natures. The results shown in Figure 6(b) are further elaborated by showing how it evidences our claim. The range of cosine similarity from 0.99 to 1 is not as relevant, since the absolute scale of similarities is not emphasized here, as the qualitative difference between inter-/intra- architecture score similarity differences.
> >
> > ---
> >
> > ## Weakness 7 /  Requested Change 4: Perfect Accuracy in Fingerprint Task and ResNet-50 Comparison
> >
> > **Comment:**
> > The reviewer noted that the perfect accuracy achieved in the fingerprint identification task raises concerns about potential data leakage or task simplicity. Additionally, they pointed out that ResNet-50 achieves similar accuracy with reduced runtime.
> >
> > **Response:**
> > We understand the concerns about perfect accuracy and potential task simplicity. The high accuracy reflects the robustness of the fingerprinting task, which is designed to differentiate models based on their embeddings. Additionally, the varying performance of models like ViT and Vanilla CNN demonstrates that the task is non-trivial, as not all models achieve the same level of accuracy, further validating the complexity of the problem.
> > ResNet50 is a pre-trained model on ImageNet, which makes it well-suited for datasets similar to ImageNet, such as CIFAR-10. However, for datasets that deviate significantly from ImageNet in terms of distribution and content, ResNet50’s performance may degrade. In contrast, our method is unaffected by changes in dataset characteristics and can be generalized better in such cases.
> > While our method may be slower, its advantage lies in being less dependent on the pre-training dataset and showing greater potential for more general fingerprinting tasks. We will include this explanation in the revised manuscript.
> >
> >
> > ---
> >
> > We hope these clarifications address the reviewer’s concerns. Thank you again for your valuable feedback.

---

### Review · Reviewer_LQjL · 2024-08-27

**Summary Of Contributions:**

This paper explores how the CT -score of encoders changes with the number of layers. It is observed that for generative models, lower layers encode low-level details of images while higher layers encode high-level semantics of images. Based on this fact, the paper proposes a method that determines whether two models are the same using the embeddings produced by their encoders.

**Audience:**

Yes

**Broader Impact Concerns:**

I have no broader impact concerns for the paper.

**Claims And Evidence:**

Yes

**Requested Changes:**

Please refer to the weakness part and make the following changes.

R1: Explain how the CT-score can be used to detect data memorization and compare it with other scores to explain its advantages.

R2: Justifying why detecting the model architecture is important, and why it is challenging to detect the model architecture when we can access the embeddings of all encoder layers.

R3: Explain the advantage of the proposed method over ResNet50.

**Strengths And Weaknesses:**

S1: Detecting data memorization is important for protecting data privacy.

S2: The experiments are comprehensive.


Weakness

W1: The content does not match the title. The title says “detecting memorization”, and one would expect that the CT-score will be used to detect memorization. However, this is not conducted.

W2: To establish that the CT-score is good method to detect memorization, a comprehensive comparison between it and other scores is required.

W3: Seems that fingerprinting refers to deciding the architecture of the generative model? It needs to be justified why the architecture of the model needs to be decided when we have all layer embeddings of the model. Moreover, why deciding the model architecture is an important problem.

W4: The proposed fingerprinting method matches ResNet50 in accuracy but is slower.

---

> ### Author Response · Authors · 2024-10-03
> **Response to Reviewer LQjL**
>
> We thank the reviewer for the valuable feedback. In response to the concerns raised, we have addressed the points below:
>
> ---
>
> ## Weakness 1 & 2 / Requested Change 1: Title and Memorization Detection
>
> **Comment:**
> The reviewer suggests that the title of the paper may lead to the expectation that our primary goal is to justify the use of CT-score for detecting data memorization, and a comparison between CT-score and other methods should be conducted.
>
> **Response:**
> We understand the reviewer’s concern, and it seems there may be some confusion regarding the focus of our work. The primary objective of our paper is not to improve or justify the CT-score as a method for detecting memorization but rather to explore its behavior across different encoder layers.
> We do not aim to establish CT-score as the best method for detecting memorization. Instead, our focus is on understanding how CT-scores vary across layers in different models and how these variations can reflect data-copying of different levels and can further be leveraged for model fingerprinting. We will clarify this distinction in the paper to avoid further misunderstandings.
>
> ---
>
> ## Weakness 3 / Requested Change 2: Fingerprinting and Model Architecture
>
> **Comment:**
> The reviewer raised concerns about the necessity of identifying model architectures when embeddings from all encoder layers are accessible, and questioned the importance of this problem.
>
> **Response:**
> The embeddings used in our fingerprinting method are from the encoder models rather than the generative models themselves. The aim of the fingerprinting method is to determine the architecture of the generative models, for which we can only access their generated images. This setting is meaningful for several downstream tasks, including model verification, security, and intellectual property protection.
>
> ---
>
> ## Weakness 4 / Requested Change 3: Comparison with ResNet50
>
> **Comment:**
> The reviewer pointed out that our method matches ResNet50 in accuracy but is slower.
>
> **Response:**
> We appreciate this observation and would like to provide additional context. ResNet50 is a pre-trained model on ImageNet, which makes it well-suited for datasets similar to ImageNet, such as CIFAR-10. However, for datasets that deviate significantly from ImageNet in terms of distribution and content, ResNet50’s performance may degrade. In contrast, our method is unaffected by changes in dataset characteristics and can be generalized better in such cases.
> While our method may be slower, its advantage lies in being less dependent on the pre-training dataset and showing greater potential for more general fingerprinting tasks. We will include this explanation in the revised manuscript.
>
> ---
>
> We hope these clarifications address the reviewer’s concerns. Thank you again for your valuable feedback.

---

> ### Comment · Reviewer_LQjL · 2024-10-04
> **Comments of the author response**
>
> (1)	Does this paper study data memorization or not? If yes, the paper uses CT-score, it needs to be shown that CT-score outperforms alternative scores. If not (which seems the case, as the experiment results are about model fingerprinting), please remove memorization from the title.
>
> (2)	Please provide some references on the applications of model structure detection.
>
> (3)	Are there specific experiment results to show the shortcomings of ResNet50 and the advantages of the proposed model?

---

### Review · Reviewer_MdyG · 2024-09-19

**Summary Of Contributions:**

## Contribution:
1. This paper proposes the  lack of justification to the layer choice underscores a pressing gap in the existing methodologies, highlighting the need for a systematic exploration to identify the most effective layers for capturing memorization phenomena accurately. It  aims to bridge this gap by conducting a comprehensive investigation into the efficacy of memorization measurement using embeddings from different layers of CNNs and transformers.
2. In this research we study CT-score, an embedding-based data-copying metric proposed in [1], to measure model memorization with ViT layer embeddings.
3. Beyond merely selecting the best layers for memorization detection, we introduce and validate a novel approach for model fingerprinting that leverages the unique distribution of the CT -scores layer-wise.

[1] Casey Meehan, Kamalika Chaudhuri, and Sanjoy Dasgupta. A non-parametric test to detect data-copying in generative models, 2020.

**Audience:**

Yes

**Broader Impact Concerns:**

Ethical concerns has been discussed in conclusion section of the paper.

**Claims And Evidence:**

Yes

**Requested Changes:**

1. In the figure 1 and 2, altought the number of layers are different for each achitecture, the $C_T$ scores/y-axis are consistent, it would be easy to follow if their range can be the same.
2. The variations of models proposed in Figure 1 and 2 is not suiffient.
3. In Section 4.2, it aims to make conclusion on the $C_T$ score on curated datasets to prove the hypothesis but the analysis on the experiments are not suifficient.
4. In most of experiments such as result in Figure 1,2 and 7, the score value suddenly drop in the last layer, would the author has a idea on the existence of it?

**Strengths And Weaknesses:**

# Strengths
1. The motivation and problems trying to be addressed proposed in this paper are insightful and novel which can be meaningful for the community.
2.  The design of the paper is well-organized and the use of $C_T$ is reasonable.
3. The author conducts the experiments on various architectures, in Figure 1 and 2, the paper visualized the ViT encoder trend based on DDPM and GAN-generated Image resectively which is clear and insightful.
4. The hypothesis proposed in the Section 3.2 is interesting where the it aims to explain the funcitionality of different layers and provides different experiments analyzing this hypothesis.
5. The propose of $C_T$-Layer Method is interesting.

# Weakness
1. Figure 1 and 2, the paper visualized the ViT encoder trend based on DDPM and GAN-generated images, however, basides DDPM and GAN, VAE is another generative model which used to be widely used. Would there is a reason why not VAE?
2. For backbones in Figure 1 and 2, there are much more other models (much more than 3 models proposed in the paper), the experiments seems not suifficient.
3. Noteably, I see there is a drop of the $C_T$ score in the last layer in Figure 1 and 2, would there possiblely a explanation on it?
4. In Figure 5, the y-axis are not the same and I think it would be better if these lines can be combined in single figure to see the trend.

---

> ### Author Response · Authors · 2024-10-03
> **Response to Reviewer MdyG**
>
> We thank the reviewer for the valuable feedback. We are glad you find our work insightful and novel. In response to your suggestions, we have addressed the points below.
>
> ---
>
> ## Weakness 1: Include VAE for comparison in Figures 1 and 2
>
> **Comment:**
> The reviewer requested the inclusion of Variational Autoencoders (VAEs) in addition to DDPM and GAN in Figures 1 and 2 to enhance comparisons.
>
> **Response:**
> Thank you for this suggestion. We believe that including VAEs is not necessary for this particular analysis, as our focus is on models that exhibit data-copying tendencies, such as GANs and DDPMs. While VAE is an important class of generative models, there are many other generative models that could be considered, but exhaustively comparing all of them is beyond the scope of this work.
>
> ---
>
> ## Weakness 2 / Requested Change 2: Add more model backbones for comparison
>
> **Comment:**
> The reviewer mentioned that the use of only three model backbones may be insufficient to demonstrate the generalizability of our results.
>
> **Response:**
> Thank you for pointing this out. We acknowledge the limited number of model backbones in Figures 1 and 2. However, as our work focuses on the trend of ViT layer-wise performance, we chose a small, representative set of backbones to avoid overcrowding the analysis. In Figure 3, we have included variations of ViT backbones to further explore these trends, demonstrating the consistency of our results across different architectures. We believe these additions provide sufficient validation for the generalizability of our conclusions.
>
> ---
>
> ## Weakness 3 / Requested Change 3: Explain the drop in CT-score in the last layer (Figures 1 and 2)
>
> **Comment:**
> The reviewer asked for an explanation regarding the drop in CT-score in the final layer across models in Figures 1 and 2.
>
> **Response:**
> Thank you for raising this observation. The drop is related to the nature of the last layer in most ViT models, where embeddings are directly aligned with classification labels, reducing their distance in the hidden space. This suggests that, regardless of the differences in intermediate layer representations, the prediction scores of the last layer embedding are close in relative distance. This convergence of embeddings in the final layer likely accounts for the drop in CT-score. We will add this explanation in the revised manuscript to clarify this point.
>
> ---
>
> ## Weakness 4: Combine the y-axes in Figure 5 for clearer comparison
>
> **Comment:**
> The reviewer suggested combining the y-axes in Figure 5 for clearer trend comparison.
>
> **Response:**
> We appreciate this suggestion. However, the key focus of Figure 5 is on the relative trend shifts between layers, which are more apparent when each model's scores are shown with their respective range. As we discussed in Section 4.1, the deliberate modification of the CIFAR-10 dataset leads to significant differences in the CT-scores, particularly indicating underfitting (high positive values). Despite these variations in absolute values, the trends remain insightful for comparing the shifts from the baseline.
>
> ---
>
> ## Requested Change 1: Adjust the range of the y-axis for Figures 1 and 2 for consistency
>
> **Comment:**
> The reviewer requested that Figures 1 and 2 be adjusted so that the y-axis ranges are consistent across models.
>
> **Response:**
> We acknowledge this suggestion. It’s more visually appealing to have the three curves in one plot, instead of three, by having a consistent y-axis range. We have modified it accordingly.
>
> ---
>
> ## Requested Change 3: Expand the analysis in Section 4.2
>
> **Comment:**
> The reviewer suggested that the analysis in Section 4.2 is not sufficiently detailed.
>
> **Response:**
> Thank you for your feedback. We will expand the discussion in Section 4.2 by including additional explanations to further support our hypothesis regarding CT-scores on curated datasets. We will also provide more analysis on the potential causes and implications of the observed trends, addressing the concerns raised.
>
> ---
>
> We hope these clarifications address the reviewer’s concerns. Thank you again for your valuable feedback.

---

### Comment · Action_Editor_Hf7N · 2024-10-03
**Review Invitation: Rebuttal and Revision for TMLR Paper 3093**

Dear Reviewers,

We invite you to review the submitted rebuttal and participate in the discussion to ensure a thorough evaluation. The authors have uploaded a revision, so please check if it addresses your initial concerns.

Thank you for your contributions!

Warm regards,

TMLR Paper 3093 Action Editors

---

### Decision · Action_Editor_Hf7N · 2024-11-18

**Recommendation:** Reject

**Comment:**

Thanks for providing the response and the revised version. While some concerns have been addressed, this submission still contains two major issues. This work addresses an important problem: detecting data memorization to protect data privacy. To this end, the authors propose using the CT-score, an embedding-based data-copying metric introduced in [a], to measure model memorization through Vision Transformer (ViT) layer embeddings. However, as noted by all reviewers, the evaluation settings are not very convincing.

[a] A non-parametric test to detect data-copying in generative models

- The reliance on CIFAR-10 and manually created samples is insufficient to demonstrate the method’s ability to effectively detect memorization in generative models. While the authors argue that their focus is on exploring the behavior of the CT-score across encoder layers rather than validating it as a detection method, this does not align with the title and abstract, which emphasize “detecting memorization.” The lack of robust experimental support raises significant concerns.


- Additionally, the fingerprinting task, described as identifying the architecture of generative models, lacks clear significance, as noted by Reviewer LQjL. The task itself appears inherently simple and does not offer substantial novelty, as highlighted by Reviewer Sf4b.

Given these significant concerns, the paper does not make a strong case for the impact of its contributions. Addressing these key issues is essential for the work to meet the standards required for publication. Please consider revising the paper to address these two major points.

**Audience:**

Yes, the paper’s findings on memorization and its new method for identifying models are useful for TMLR’s audience, especially those interested in AI security and misinformation.

**Claims And Evidence:**

This paper investigates data memorization in generative models, focusing on memorization scores derived from Vision Transformer (ViT) embeddings. The authors observe a layer-specific trend where deeper layers exhibit reduced memorization and propose that early layers capture low-level features, while deeper layers reflect high-level semantics. Building on this insight, they introduce a fingerprinting methodology based on layer-wise memorization score distributions.

- As indicted by all reviewers, the evaluation is not very convincing. The experimental settings are not extensive, as the evaluation is soly on CIFAR-10. Reviewer Sf4b also pointed out fingerprint identification task is simple.

**Resubmission Of Major Revision:**

The authors may consider submitting a major revision at a later time.